# Improved Tensile and Bond Properties through Novel Rod Constructions Based on the Braiding Technique for Non-Metallic Concrete Reinforcements

**DOI:** 10.3390/ma16062459

**Published:** 2023-03-20

**Authors:** Anwar Abdkader, Paul Penzel, Danny Friese, Matthias Overberg, Lars Hahn, Marko Butler, Viktor Mechtcherine, Chokri Cherif

**Affiliations:** 1Institute of Textile Machinery and High Performance Material Technology (ITM), Technische Universität Dresden, 01062 Dresden, Germany; 2Institute of Construction Materials, Technische Universität Dresden, 01062 Dresden, Germany

**Keywords:** carbon-reinforced concrete, bond behavior, tensile test, braided rods, bond test, profiled roving, non-metallic reinforcement

## Abstract

Textile reinforcements have established themselves as a convincing alternative to conventional steel reinforcements in the building industry. In contrast to ribbed steel bars that ensure a stable mechanical interlock with concrete (form fit), the bonding force of smooth carbon rovings has so far been transmitted primarily by an adhesive bonding with the concrete matrix (material fit). However, this material fit does not enable the efficient use of the mechanical load capacity of the textile reinforcement. Solutions involving surface-profiled rods promise significant improvements in the bonding behavior by creating an additional mechanical interlock with the concrete matrix. An initial analysis was carried out to determine the effect of a braided rod geometry on the bonding behavior. For this purpose, novel braided rods with defined surface profiling consisting of several carbon filament yarns were developed and characterized in their tensile and bond properties. Further fundamental examinations to determine the influence of the impregnation as well as the application of a pre-tension during its consolidation in order to minimize the rod elongation under load were carried out. The investigations showed a high potential of the impregnated surface-profiled braided rods for a highly efficient application in concrete reinforcements. Hereby, a complete impregnation of the rod with a stiff polymer improved the tensile and bonding properties significantly. Compared to unprofiled reinforcement structures, the specific bonding stress could be increased up to 500% due to the strong form-fit effect of the braided rods while maintaining the high tensile properties.

## 1. Introduction

In civil engineering, carbon-reinforced concrete (CRC) is of great interest to both research and industry in order to decrease the tremendous CO_2_ emissions of up to 38% globally in the building industry significantly [1,2]. Due to corrosion resistance and a high load-bearing capacity compared to conventional steel bars, CRC allows up to 80% reduced concrete overlay and a lifespan of more than 200 years, making it a perfect material for a sustainable building of the future [3]. It is especially suitable in the use of thin-walled and lightweight concrete components with high load-bearing capacities as well as for reinforcing existing constructions [4].

In order to exploit the excellent properties of CRC, the bonding between the textile reinforcement and the surrounding concrete matrix (outer bond) as well as the bonding between the single filaments (internal bond) is of great importance. Especially when considering the energy-intensive production of the carbon fiber textiles, a high material utilization is needed for an efficient and resource-saving application of CRC for a sustainable building of the future. 

In accordance with this, the bond mechanisms in textile reinforcements have long been the subject of extensive research [5,6,7,8,9,10]. A major prerequisite for the effective material utilization of the reinforcement potential in concrete is a strong internal bonding, since otherwise the core filaments of the rod will be significantly less or not at all involved in the load transmission, and the bond will fail accordingly early [5,9]. Furthermore, the strain (ε) at tensile failure of the reinforcement rod must be limited (ε ≤ 0.2%) in order to transmit the full load at small deformations of reinforced concrete structures and in order to ensure small crack widths in the composite as soon as the concrete matrix fails under tensile stress at app. 0.2% strain. Increased structural elongation of the reinforcement rod results in excessively wide cracks (bad durability) or high deflections (bad functionality) [7]. Therefore, a suitable rod structure (surface profiling) and consolidation as well as complete impregnation must ensure a good internal bonding [8].

As a result, various impregnations were developed to improve the internal bonding between the filaments [9]. Impregnations for rovings commonly used today are based on styrene-butadiene rubber (SBR), epoxy resin (EP), vinyl ester resin (VE), or polyacrylate (PA) [8,10]. All of these improve the internal bonding compared to unimpregnated rods and improve the structural stability as well [9]. SBR-impregnated reinforcements are more flexible and are therefore suitable for curved shapes and thus for retrofitting of existing buildings [11], whereas stiff EP-impregnated reinforcement structures are preferably used for precast concrete structures [12].

The outer bond between the rod and concrete matrix is based primarily on three mechanisms as shown in Figure 1 [13,14,15,16]. The Adhesive bond is based on adhesion (chemical bonding) between impregnation and hydrated cement. A relative displacement between the rod and the concrete overcomes the adhesion and activates the frictional bond. It primarily depends on the roughness in the boundary layer between rod and concrete matrix [15,17]. The highest outer bond is achieved through a form-fit-based shear bond (mechanical interlock) between a profiled rod structure and the surrounding concrete.

The first studies with continuously profiled carbon rovings showed that the bonding behavior could be adapted by surface profiling and variations in the profile depth (form fit) [18]; however, the exact mechanisms and principles of mechanical interlocking between the rod and concrete through profiling remain to be researched. For the analysis of the bond mechanisms, the bond behavior is characterized using the derived pull-out force–slip deformation relationships or the derived bond stress–crack-opening relationship [14,19].

An approach to increasing the bonding behavior of reinforcement yarns is the profiling of carbon rovings through a shaping process that was developed at the Institute of Textile Machinery and High Performance Material Technology (ITM). Hereby, the patented tetrahedral profile [20] is created by shaping the freshly polymer-impregnated roving with interlocking profile tools. The profiled yarn/roving is permanently stabilized through the cross-linking of the impregnation polymer via infrared (IR) radiation during the shaping process. This method allows the production of profiled rovings with different profile configurations. They show different and well defined bonding behaviors depending on the profile parameters [18,21].

Reinforcement structures with a large cross section compared to grid-like textile reinforcements can be produced by pultrusion from a single-fiber system [22]. Using polymeric matrix material in combination with thermal processes, fiber-reinforced plastics (FRP) profiled along the yarn axis can be generated by embossing [22]. The milling of rib-like structures into pultruded reinforcement bars (rebars) has also become a sufficient solution to produce rigid rebars with, in some cases, very high and reproducible bond parameters [23]. However, the methods are not suitable for reinforcements with smaller yarn cross-sections such as those required in flat components or for building reinforcements in combination with a low concrete layer thicknesses. Furthermore, the separated filament course of the milled structure results in shearing of the ribs and telescopic pull-out failures of concrete embedded rebars [24] in addition to a reduced efficiency with regard to material use and resource savings. 

Other established and well-known methods for creating undulated yarn architectures in favor of a profiled surface and reproducible characteristic are, for example, braiding, twisting, and cabling [25]. With these technologies, the formation of a defined yarn geometry is realized by alternately inserting, intertwining, or winding of the individual threads. As a result, an improvement in the internal fiber bond and outer bond with the surrounding concrete matrix connection to the yarn surface is achieved [26].

This paper presents braiding technology for the development of novel rod structures from carbon rovings with defined yarn properties; e.g., a defined force–strain behavior; a stretched, shear-resistant anchored yarn structure; and surface profiling. The reinforcement structures made from these rods are expected to have significantly improved shear stress transmission in concrete with adjustable property profiles (Young’s modulus, structural elongation, bond stress, and tensile strength) in combination with a high tensile strength.

The following Table 1 gives a brief overview of critical work and the state of the art this paper was based on.

## 2. Materials and Methods

The focus of the study was the characterization of tensile and bond properties of different braided rods in comparison to unprofiled carbon rovings. In order to reduce the structural elongation of the braided rods, a purposeful impregnation and subsequent consolidation was investigated. Hereby, the influence of different impregnation agents and the solid content of the impregnation was studied to derive process recommendations. In the case of braided rods with a very high rod count and therefore a high predicted structural elongation, a pre-tension force was applied during the consolidation process in order to stretch out the rod and stabilize this condition for a further reduction in the structural elongation (improvement in the Young’s modulus). For the characterization of the braided rods, tensile and pull-out test are performed. Furthermore, micrographic analyses were conducted to evaluate the impregnation quality and rod geometry.

### 2.1. Braided Rods Composed of Carbon Fiber Heavy Tows

For a broad characterization of the tensile and bonding properties of braided rods, different reinforcement rods consisting of several carbon fiber heavy tows (CFHT) from the Teijin Carbon Europe GmbH (Wuppertal, Germany) [29] were produced. An overview of the used fiber material is shown in Table 2. The tensile properties were characterized according to ISO 3341 [30] (Section 3.3) at the ITM. The tensile strength and Young’s modulus refer to the filament area, which is determined by the density and count of the CFHT (for 3200 tex CFHT: 1.81 mm²).

A 4 × 4 variation braider VF 1/4-32-140 from Herzog GmbH (Oldenburg, Germany) was used as a basic braiding machine for targeted further development of the machine technology and for the development of braided rods (Figure 2).

The specifications of the variation braider were:4 × 4 impellers to “park” several bobbins;Central control of the impellers;Up to 32 clappers and 24 pneumatic switching points;Clappers with closed ceramic yarn guides;9 core inlet tubes Ø 8 mm between the impellers for core yarns;16 drilled paddle wheel pillars for standing threads.

The variation braider was specifically further developed for gentle processing of brittle fibers (e.g., carbon filament yarn) with regard to the following points:Optimization of the bobbin design;Optimization of the deflection rollers with regard to surface properties and geometry;Optimization of the yarn tension compensation.

Using the modified VF 1/4-32-140 variation braider, continuous braided rods with different surface profiles were produced according to Table 3 with a 4-strand flat braid composed of 800 tex CFHTs (total rod count: 3200 tex) and a 6-strand flat braid composed of 3200 tex CFHTs (total rod count: 19,200 tex).

### 2.2. Impregnation and Consolidation

In order to increase the internal bonding and for structural fixation, the produced braided rods were impregnated in an immersion bath method with water-based polymeric dispersions and consolidated using direct IR radiation. For the investigation, different impregnation agents were used (Table 4). 

In a series of experiments, the resulting solid polymer content in the impregnated rods was varied (Table 4) by adding normal water to the polymeric dispersion. For the consolidation (drying and stabilization), the impregnated rods were positioned between two opposite positioned IR modules from Heraeus Noblelight GmbH (Kleinostheim, Germany) with the specification Typ MW Gold B 9755255 2500 W (230 V) with fast middle wave and 50% power (1.25 kW) and a distance of 10 mm to the roving. To minimize structural elongation, the braided rods were subjected to different pre-tension forces during the impregnation and consolidation process. 

As a reference structure for the braided rods with a low count (3200 tex), an impregnated straight roving (with a circular cross section) consisting of a single 3200 tex CFHT was produced (called impregnated roving).

In the case of braided rods with a very high rod count and therefore a high predicted structural elongation, a pre-extension force was applied during the consolidation process in order to stretch out the rod and stabilize this condition for a further reduction in the structural elongation (improvement in the Young’s modulus). The levels of the applied pre-extension force in order to reduce structural elongation and increase the Young’s modulus were determined in tensile tests of dry braided rods and are further justified in Section 3.1.

In order to apply the pre-extension force during consolidation for minimizing the structural elongation of the braided rods with a high count, a test stand consisting of a frame, two opposite IR radiators, and a suspension was used. The schematic setup for the production of the samples composed of pre-extended braided rods with a high count is shown in Figure 3.

An overview of all test series of the different reinforcing rod architectures is given in Table 5.

The three different investigated braided rods for concrete reinforcement are illustrated in Table 6.

### 2.3. Concrete Matrix

Fiber-based reinforcements are very often embedded in cementitious matrices with a small maximum grain size of 1–2 mm [6].

For the characterization of the bond and tensile behaviors of concrete-embedded braided rods, two different fine concretes were used. The used high-performance concrete for the pull-out tests consisted of a fine concrete dry-mix called BMK 45-220-2. The detailed composition is presented in [21].

For the tensile test on carbon-reinforced concrete (CRC) specimens, the fine concrete TF 10 CARBOrefit^®^ was used; it consisted of the fine concrete dry mix TF10 by the company PAGEL Spezial-Beton GmbH & Co. KG (Essen, Germany). The plastic consistency of concrete is suitable for laminating in layers and therefore for production of sample plates of CRC tensile test specimens.

The basic mechanical properties of the concrete were determined on prisms (dimensions 40 × 40 × 160 mm³) according to DIN EN 196-1 [33]; the mean values are listed in Table 7.

### 2.4. Characterization of the Rods’ Tensile Properties and Bond Behavior in Concrete as Well as Tensile Properties of the Composite

For the determination of the tensile properties, the different dry and impregnated braided rods were tested at the Institute of Textile Machinery and High Performance Material Technology (ITM), TU Dresden. The dry braided rods were tested according to ISO 3341 [30] with wrap clamps. Due to the stiffness of the impregnated braided rods, which resulted in filament damage when wrapped around the clamps, the ends of the rods were resinated, clamped by metal clamps with a free clamping length of 200 mm, and tested acc. to DIN EN ISO 10618 [34]. The detailed test setup and testing parameters are presented in [18,21]. All tensile tests were performed with a Zwick Z100 testing machine. The tests were performed in a normal climate according to DIN EN ISO 139. A minimum of 10 specimens for each series were tested. 

In order to determine the quality of the impregnation and the cross-sectional geometry of the different impregnated reinforcement rods, a micrographic analysis was conducted. Hereby, resin-embedded samples of the reinforcement rods were cut through the circumference and were analyzed with the AXIOImager.M1m microscopic workstation by the company Carl Zeiss AG (Germany) at the Institute of Textile Machinery and High Performance Material Technology (ITM), TU Dresden, with a magnification factor of 200:1 in a bright field and a reflective light microscopy method. 

For the computer-supported analysis of the impregnation agent amount, its distribution, and the presence of air voids, the open source image-processing software ImageJ Version 1.54b was used to color the different components (filaments, polymer, embedment resin, and air gaps) by assigning new colors to detected color ranges. Hereby, the picture was transformed into a four-color image with one assigned color for each component. Once calibrated by manual selection of the color values for each component, the software allowed batch processing for similar images. The open source software ilastik enabled the image classification and segmentation of the colored images by counting the different pixel values of the colored microsection in a histogram. The diameter of the cross-section was determined by manual evaluation of the microsection dimension. In case of elliptical deformed geometries, its major axis and minor axis (the longest and shortest dimension) were measured and averaged.

For the tensile behavior composite composed of concrete and embedded braided rods with low fineness, tensile tests of composite specimens (reinforcement rod and concrete matrix) were performed at the Institute of Concrete Structures (IMB), TU Dresden. All tests took place at 20 °C exactly 28 days after casting according to the recommendations in [15]. Detailed information about the test setup and specimen production is given in [21]. Three composite samples (each reinforced by three braided rods) were tested.

For the characterization of the bonding behavior of the single reinforcement rod, pull-out tests acc. to [28] were conducted at the Institute of Construction Materials (IfB), TU Dresden. The specimen for the pull-out testing consisted of an upper concrete block to cause the pull-out of the reinforcement rod with an embedding length of 50 mm (bond length) and a lower concrete block for the rod fixation with an anchoring length of 90 mm. The test setup and specimen production are described in [21]. 

The test setup comprised a tensile strength testing machine by Instron 8802 with one upper, smaller specimen holder composed of metal and one lower specimen holder (Figure 4). The specimens had a free length of 120 mm. The test velocity was 1 mm per minute. The concrete-embedded specimens were tested at 20 °C, 65% relative humidity, and a standard ambient pressure of 1 bar. At least five specimens for each series were tested. 

### 2.5. Specimens Manufacturing

For the microscopic examinations, 10 mm-long roving sections were placed in cylinders with a 20 mm diameter and fully resinated. After one day of drying, the front side was ground with sandpaper several times from rough to fine and finally polished with a polishing agent.

For the rod tensile tests, 450 mm-long rod sections were cut to size. Then, the rods were stretched and clamped in a frame. With the help of metal molds, the ends were embedded in epoxy resin at 125 mm each. 

For the determination of the tensile strength of rods embedded in TF10 CARBOrefit© fine concrete, six tensile specimens with three profiled rods each and a concrete cover of 5 mm were produced for each rod series by laminating. This was done in a formwork in which the individual rods were fixed and aligned with a rod spacing of 13 mm. Then, the fine-grained concrete was filled in; first a bottom layer was created that was subsequently slightly compressed. In a second step, the top concrete layer was filled in and smoothed. The 120 cm-long, 1 cm-thick, and 33 cm-wide plate was then covered with damp cloths. From the 2nd to 7th days, the plate was stored in water. From day 8 to day 28, the plate was stored in a climate chamber at 20 °C and 65% relative humidity. Before the tensile tests, the plate was sawn into 5.2 cm-wide strips containing three rods each.

Specimens for the yarn pull-out (YPO) tests were created by embedding single braided rods as well as rovings with no profile (reference) in self-compacting, fine-grained concrete (BMK 45-220-2) in a cube formwork. The specimens consisted of two centered concrete blocks at the rod ends and a free rod segment of 120 mm in between the blocks. This clearly defined the area in which composite failure could occur. The specimens were stored for 7 days under water and stored for additional 21 days in a climate chamber (20 °C and 65% relative humidity).

## 3. Results and Discussions

### 3.1. Tensile Properties

The diagrams in Figure 5 and Figure 6 illustrate the mean values of the tensile test results of the different series of reinforcement rod configurations with their standard deviations. For each of the following series, at least 10 single specimens were tested according to DIN EN ISO 10618 [34] (impregnated rods) and ISO 3341 [30] (dry braided rods). In order to analyze the influence of the braiding as well as the impregnation parameters, an important part of the study was a comparison between the tensile properties of dry braided rods (4_800, compare Table 5), impregnated braided rods with different impregnation agents (4_800_SBR, 4_800_PA), and impregnated rovings with a circular geometry (1_3200_PA) as well as impregnated braided rods with reduced polymer content (4_800_X).

The determined tensile strength (in N/mm²) refers in all tests (dry and impregnated rods) to the measured force (in N) divided by the net filament area (actual area of reinforcement). For the 3200 tex single roving and the braided rods with a total rod count of 3200 tex (4 × 800 tex), the compact filament area was 1.81 mm². The braided rods with a high fineness of 6 × 3200 tex had a compact filament area of 10.86 mm² (6 × 1.81 mm²). 

The following Figure 5 shows the influence of a varied polymer content of the rod on the tensile properties of 4 × 800 tex PA-impregnated braided rods.

The study’s results showed that a reduced polymer content in the impregnated rod resulted in reduced tensile properties (tensile strength and Young’s modulus). The braided rods with 22% mass content of the impregnation (4_800_PA) showed a result of 3524 MPa, which was the highest tensile strength, as well as the highest Young’s modulus at 211 GPa in addition to the smallest standard deviation, indicating a good and even impregnation quality (complete impregnation) and an improved internal bond between the filaments that resulted in a more even load distribution. Furthermore, the tensile properties were in the range of approved textile reinforcements for concrete applications according to the general building approval [27] (values according to [27]: tensile strength ≥2700 MPa; Young’s modulus ≥170 GPa) with an additional safe limit of up to 20%. Further reducing factors were regarded in the design of the reinforcement structure as described in [27]. 

With reduced mass content of the impregnation down to 10 M.-% (4_800_PA_10), the tensile properties dropped almost 25%, and the standard deviation increased significantly to almost ±20%. As deduced from that, the reduced solid content of the impregnation resulted in an uneven and incomplete impregnation of the braided rods and therefore an uneven load distribution between the filaments when stressed. The reduced Young’s modulus especially indicated an increased structural elongation under load. The thesis was that a higher impregnation mass-content impregnation and therefore a higher polymer content in the braided rod resulted in a polymer accumulation in the gaps of the braided structure between the single fibers and reduced or limited the transverse contraction of the braided rod under load depending on the polymer proportion and stiffness, resulting in a higher tensile stiffness (Young’s modulus) and a reduced structural elongation.

Figure 6 shows the mean values of the tensile properties with a single standard deviation of four different braided rods with a total count of 3200 tex. All impregnated rods in Figure 6 had an averaged polymer content of ~22 M.-%, which was determined as favorable in order to achieve high tensile properties (see Figure 5).

The diagrams in Figure 6 show that the impregnated heavy tow roving (1_3200_PA) had the highest tensile strength at app. 3830 MPa and Young’s modulus at app. 236 GPa and thus only typical structural elongation under load. The impregnated braided rod with PA as an impregnation agent (4_800_PA) had very high tensile properties as well: an app. 10% decrease compared to the impregnated roving (3524 MPa tensile strength and 211 GPa Young’s modulus), which resulted from the slight deviation in the individual filaments in the braided rods compared to the perfectly uniaxial orientated filaments in the straight roving. Therefore, no complete uniform load distribution was possible, and lateral forces were introduced in the filaments due to the transverse contraction of the braided structure, thereby resulting in a slightly premature failure compared to the straight impregnated roving. However, no structural elongation under load was recognizable. The SBR-impregnated braided rod (4_800_SBR) had (at a 2190 MPa tensile strength and a 188 GPa Young’s modulus) significantly lower tensile properties compared to the PA-impregnated braided rod and an evident structural elongation of app. 0.3% due to the soft SBR impregnation. The lowest tensile properties show unimpregnated, dry braided rods (4_800) with a tensile strength of 1773 MPa, a Young’s modulus of 93 GPa, and a significant structural elongation under load of ca. 0.5% resulting from the undulated braided rod structure. 

The results showed that impregnation increased the tensile behavior of the braided rods significantly and that the PA impregnation was much stiffer compared to the SBR impregnation and therefore more suitable in the use of reinforcement rods with reduced structural elongation. A PA-impregnated braided rod showed no evident structural elongation under load and similar tensile properties compared to straight impregnated rovings.

Figure 7 shows the averaged force–strain behavior of the dry braided rods with a high rod count of 6 × 3200 tex as well as the linear behavior (tangent), the structural elongation under load Δε (non-linear behavior, red dotted line) and the determined pre-extension forces for stretching out of the braided rod during consolidation (dotted lines).

In contrast to the braided rods with a relatively low count of 4 × 800 tex (total rod count 3200 tex), the dry braided rods with 6 × 3200 tex (total rod count 19,200 tex) had very low tensile properties with a tensile strength at 684 MPa (6420 N breaking strength), a Young’s modulus of about 17 GPa, and a structural elongation of app. 1.2% (elongation at failure app. 2.4% composed of 1.2% structural and 1.2% material elongation under load). Compared to the braided rod with a low count, the braided rod with a high count had a 60% lower tensile strength but a 250% higher structural elongation. This inefficient use of the performance potential was due to the strongly undulated braided structure of the six braided 3200 tex CFHTs with significant voids between the single CFHTs compared to a more dense braided structure of the braided rod with a low count. The undulated structure increased the transverse contraction of the braided rod under load, resulting in higher structural elongation, lateral forces, and therefore premature failure compared to a roving with a comparable count but straight filament orientation.

Such a high structural elongation under load would be unsuitable for use as reinforcements in concrete structures because after the failure of the concrete matrix at ~0.2% strain, a further structural elongation of the reinforcement rod resulted in increased crack openings and deflection of the structure. For a reduction in the structural elongation of the impregnated braided rod with a high count, a pre-extension force was applied during consolidation (after impregnation) according to the load levels measured in the force–strain curve (1000 N/2000 N/3000 N). The thesis was that the applied load would stretch out the 1.2% structural elongation of the dry braided rod according to the load level as shown in Figure 7 and that the consolidation process stabilized the structure in the outstretched condition and therefore reduced the structural elongation under further load application. Hereby, a pre-extension force of 1000 N would reduce the structural elongation by 0.8% down to 0.4%, 2000 N would reduce elongation to 0.1%, and 3000 N would reduce elongation completely to 0%. 

The diagram in Figure 8 shows the averaged stress–strain curve of the impregnated and outstretched braided rods with a high rod count in comparison to the dry braided rod.

It was evident that the impregnation of the braided rods with a high count increased the tensile properties and reduced the structural elongation under load significantly, which was similar to the braided rods with a low count. Figure 9 compares the averaged tensile properties.

Hereby, the impregnation with the stiff PA impregnation agent and a high polymer content increased the tensile strength by almost 250% to 1634 MPa, and the Young’s modulus was at about 97 GPa (five times higher). The structural elongation was almost eliminated.

The application of a pre-extension force resulted in an up to 10% increase in the Young’s modulus (ca. 110 GPa) and a similar tensile strength at around 1574 MPa compared to the impregnated braided rods. Therefore, the tensile stiffness was increased and the structural elongation could be reduced. Higher pre-extension forces resulted in a slightly lower tensile strength at 1420–1470 MPa and an increased single standard deviations with up to 15% (probably due to increased fiber damage) but a higher Young’s modulus at ~110 GPa. 

In summary, the investigations showed that an impregnation with high polymer content of at least 22 mass-% (see Figure 5) and a stiff impregnation agent on a PA basis (see Figure 6) were favorable to achieve the highest tensile properties of braided rods. In the case of braided rods with a high rod count (e.g., 6 × 3200 tex), the application of a limited pre-extension force during consolidation increased the Young’s modulus (see Figure 9), making them suitable for transmission of high initial forces with reduced structural elongation.

### 3.2. Micrographic Analysis

In order to validate the quality of the impregnation, the ratio of the polymer content as well as to examine the dimensions of the impregnated reinforcement rods, microsection analyses were conducted as described in Section 3.4 in addition to a gravimetric analysis for the determination of the polymer content. The microscopic tests were performed at the textile physical testing laboratory at the Institute of Textile Machinery and High Performance Material Technology (ITM), TU Dresden. At least four cross-sections of each following series were analyzed. Hereby, the following colors of the original microsections indicate different components: white—filament; grey (outside)—embedment resin; light grey (inside)—impregnation/polymer; and black—air gaps (see Table 8). For a computer-supported analysis of the impregnation ratio (filament/polymer/air content), image-classification programs were used for a coloring of the components (see Section 3.3). 

Table 9 shows an exemplary comparison of an original and colored microsection of a braided rod and the component assignment.

The results of the micrographic and gravimetric analysis showed that all rod types had at ca. 22 mass.-% a similar polymer content. In contrast to the straight, impregnated roving with no evident polymer accumulations and a comparatively dense fiber arrangement, due to linear orientation of the filaments without undulation, the braided rods, which consisted of several braided filament yarns, showed an increased accumulation of the polymer in the voids between the single filaments in the internal yarn structure. Furthermore, the micrographic analysis showed minor air voids of the impregnated rod structure (≤1% for braided rods with low count) with up to 2% for braided rods with a high count. Such air voids or inclusions could destabilize the rod structure and result in reduced tensile properties.

The micrographic analysis presented in this study used a computer-supported evaluation of the quality of the impregnation (polymer content, distribution, air voids, etc.). It allowed a fast and exact investigation of the rods’ material and structural properties such as polymer accumulations and air voids as well as the dimensions and filament arrangement, which were needed to verify a high impregnation quality and for calculation of the bond stress on the basis of the rod dimensions. Especially when deriving correlations between rod structures with defined surface-profiles and the resulting tensile and bonding properties, such investigations are needed for further understanding of the inter-relations. 

### 3.3. Tensile Properties of Concrete-Embedded Braided Rods

Figure 10 shows the force–strain behavior of three individual composite test specimens, which consisted of three concrete-embedded braided rods with a low count and PA impregnation (4_800_PA) with a very low scatter. 

The evaluation of the CRC tensile test specimens is compared in Table 10.

A mean failure force of 16.9 kN or a mean rod tensile strength of 3110 MPa, referring to the dry and compacted filament cross section of 1.81 mm² for each rod, was determined. These values were in the range of the tensile strength of the single rod tensile tests (see Section 3.1) and therefore confirmed a high material utilization due to an almost uniform load distribution. According to the general building approval [27], the concrete-embedded rod required a tensile strength ≥2250 MPa. The braided rods achieved up to 40% higher tensile properties and therefore had a very high safety factor and great suitability for use as concrete reinforcement.

In all tests, rod rupture occurred, indicating a full load transfer between the textile and concrete matrix. The braided structure was still visible after testing. The rods were not pulled out from the load transfer areas, indicating a sufficient outer bond between the braided rods and the concrete matrix. After appearance of several cracks (an average of eight cracks before failure) in an evenly distributed pattern during the test, speaking for an even load distribution, the concrete was completely spalled in the measuring area at the moment of rod failure (see Table 10). During the tests, no splitting or delamination cracks could be observed. 

In conclusion, the force–strain behavior curve or the tensile strength was similar to that of the reference expansion specimens composed of construction-approved reference textiles (e.g., CARBOrefit^®^ variant 3 with 3200 tex CFHT and PA impregnation [27]).

### 3.4. Bond Properties

The following Figure 11 shows the averaged pull-out force–slip-deformation curves of the PA- and SBR-impregnated braided rods with a low count (4_800_PA/SBR) in comparison to the impregnated roving (1_3200_PA).

Due to the improved mechanical interlock of the braided rods, the resulting bond strengths were well above that of the impregnated rovings with no profile. The PA-impregnated braided rods (4_800_PA) transmitted with approximately 5.1 kN, which was almost five times higher than the pull-out forces for impregnated rovings with no profile (0.8 kN), and showed a distinct shear bond. In addition to that, they showed a steep increase in the pull-out force, emphasizing a high bond stiffness and strong anchoring (mechanical interlock and dominant shear bond). The sudden drop in the bond force was due to the filament failure at 5.1 kN, indicating that the embedment length of 50 mm was sufficient for a complete anchoring and load transmission of the braided rod in the concrete matrix. In contrast to that, the impregnated roving with no profile (1_3200_PA) was pulled out completely due to a lack of a sufficient shear bond and mainly an adhesive or frictional bond. The SBR-impregnated braided rod (4_800_SBR) initially showed a steep increase in the pull-out force due to an initial shear bond, but the mechanical interlock failed prematurely at about 1.2 kN, and the braided rod was pulled out and showed only frictional bond. A possible reason for the premature destruction of the mechanical interlock was a deformation of the braided structure due to the “soft” SBR impregnation. The results showed that the stiff PA impregnation was suitable for creating profiled reinforcement rods with a strong mechanical interlock, a distinct shear bond, and significantly increased bond performance.

Due to the different rod count and therefore the different load distribution among the filaments, the transferable pull-out force of the rods with a total count of 3200 tex were not comparable to the braided rods with a total rod count of 19,200 tex (6_3200_PA). For a direct comparison, the bond stress (in N/mm²) had to be determined by dividing the measured pull-out force by the rods’ outer surface area. The outer surface area was hereby calculated by the product of the initial bond length (50 mm) and the averaged rod circumference of the different reinforcement rods. For simplification of the complex surface geometry of the braided rods, the circumference was calculated based on a circular cross-section with the comparing diameter of the rods, which was determined in the micrographic analysis in Section 3.2 (see also [23] for the comparing diameter) multiplied by pi.

Figure 12 shows the averaged bond stress–slip deformation curves of the different reinforcement rods, in which the comparison of the standardized bond stress of the braided rods between low and high rod count can be seen.

Hereby, the braided rod with a low count (4_800_PA) and a high count (6_3200_PA) had an almost identical bond stress–slip deformation behavior. The initial bond stiffness was identical, yet the braided rod with a high count showed, at 19.3 N/mm², an almost 20% higher bond stress than braided rods with a low count, which possibly resulted from the rougher and more undulated braided structure and therefore strong mechanical interlock. In comparison with the impregnated roving without profile (2.5 N/mm²), the braided rod with a low count (4_800_PA) showed a 544% higher bond stress behavior, and the one with a high count (6_3200_PA) showed a 672% higher bond stress behavior. The bond stress of the braided rod with a low count and SBR impregnation agent (4_800_SBR) was 44% higher compared to the impregnated roving without profile.

## 4. Conclusions

The results showed that the developed braided rods were able to transmit much higher pull-out loads than common carbon fiber architectures without profile and demonstrated a significantly improved bond-slip behavior with up to five times the maximum bond strength compared to straight carbon rovings with no profile, yet maintained high tensile properties with almost no structural elongation and properties above the required values in the general building approval with sufficient safe limits, making them particularly suitable as concrete reinforcement. Hereby, the investigation of different coating agents and coating agent contents showed that a complete impregnation with a stiff polymeric impregnation resulted in very high tensile and bond properties and a strong mechanical interlock due to a reduced rod deformation under load. Braided rods with a high count showed similar results to braided rods with a low count, yet the tensile performance potential was not fully used because the high rod count resulted in an more undulated braided structure. Nonetheless, a complete impregnation eliminated the structural elongation under load, and the rough structure increased the bond behavior. As expected, the application of a pre-extension force increased the modulus of elasticity but had no significant effect on the tensile strength of the braided rods with a high count.

As a result of the bond investigations, fundamental knowledge was gained about the force transmission between the textile reinforcement and the fine concrete matrix. These are required for the subsequent design and dimensioning of the anchorage lengths of the new carbon concrete. As a major result, the braided rod designs developed were able to significantly increase pull-out forces while maintaining the concrete composite’s tensile strength.

## 5. Outlook

The developed braided rods with a low count are perfectly suitable for the use in rollable, grid-like textile reinforcement structures, whereas the braided rods with high count can be used for rigid reinforcement mats known from conventional steel reinforcements or for individual reinforcement bars used as additional reinforcement elements.

For a production of profiled, grid-like textile reinforcement structures for concrete applications in form of rollable or rigid reinforcement mats, the conventional textile manufacturing processes such as the multiaxial warp knitting process as well as robot supported rod placement will be adapted and further developed. Further research is planned on braided rods with various braiding structures for specific surface profiles and defined tensile and bond properties for application specific designs of CRC-structures and highest material efficiency. Furthermore epoxy-resin impregnated braided rods with a high rod count will be investigated for further improvement in the tensile behavior.

Such carbon fiber reinforcements with high tensile and bonding properties will clearly increase the material efficiency of carbon-reinforced concrete in the future, especially in the areas of new construction and strengthening. In the case of component strengthening, for example, shortened end anchorage and overlap lengths will improve handling. Additionally, the lower material consumption will reduce costs, which increases competitiveness compared to other reinforcement methods.

## Figures and Tables

**Figure 1 materials-16-02459-f001:**
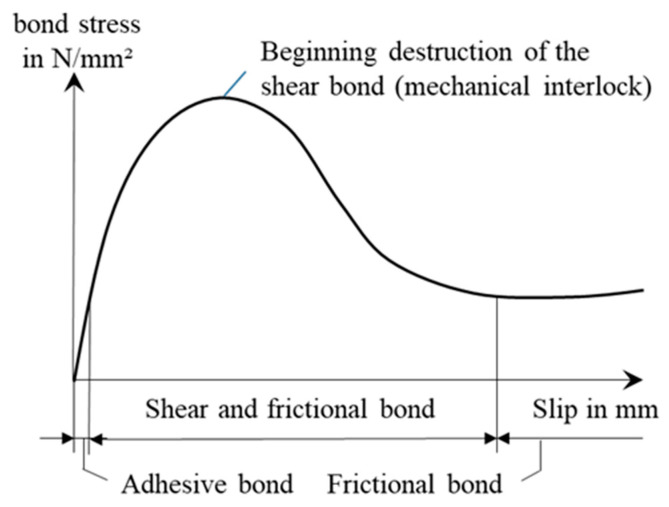
Schematic bond mechanisms of impregnated yarns for concrete reinforcements.

**Figure 2 materials-16-02459-f002:**
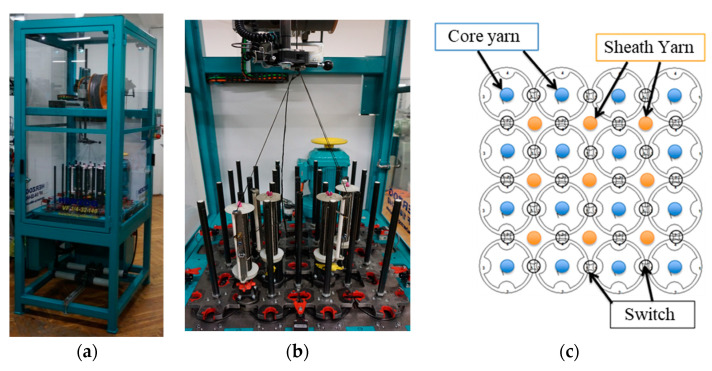
Illustration of the VF 1/4-32-140 braiding machine (**a**), the machine bed (**b**), and scheme (**c**).

**Figure 3 materials-16-02459-f003:**
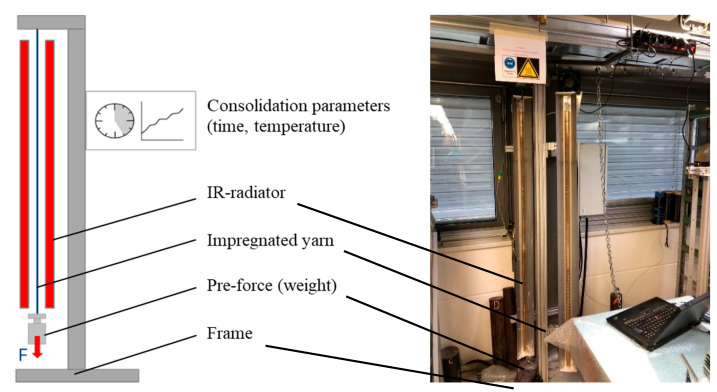
Schematic setup for consolidation with applied pre-extension force.

**Figure 4 materials-16-02459-f004:**
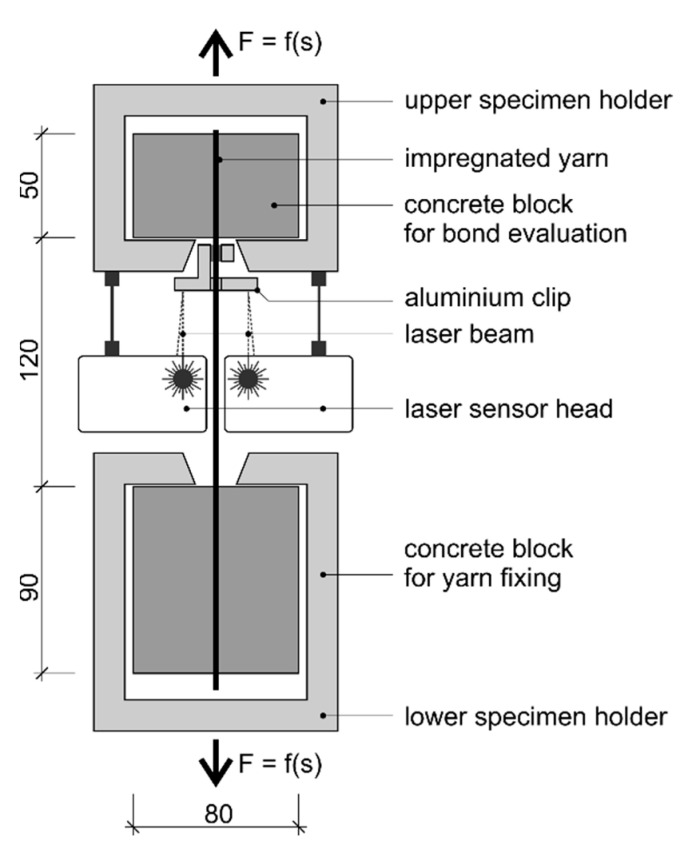
Schematic pull-out test setup of concrete-embedded braided rods (measured in mm) [8].

**Figure 5 materials-16-02459-f005:**
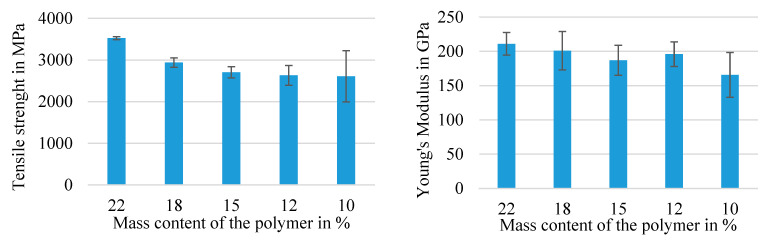
Influence of the polymer content on the tensile properties of 4 × 800 tex braided rods.

**Figure 6 materials-16-02459-f006:**
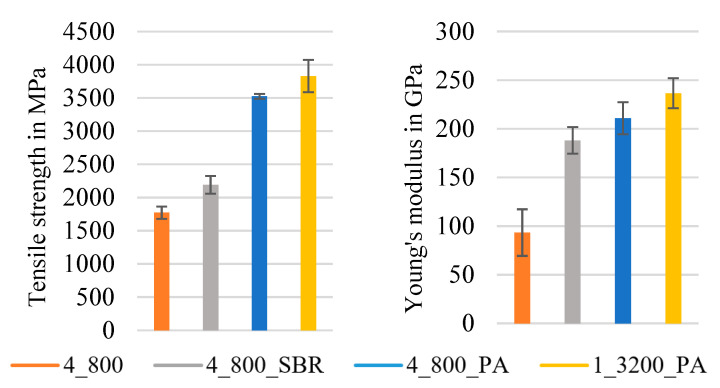
Tensile properties of impregnated 4 × 800 tex braided rods.

**Figure 7 materials-16-02459-f007:**
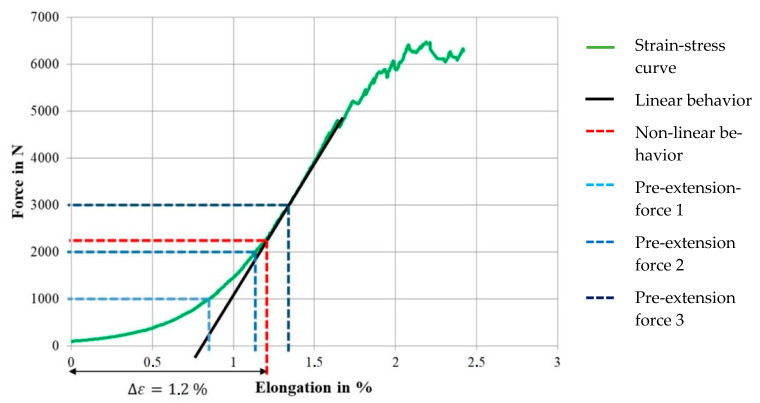
Structural elongation and pre-tension levels of dry 6 × 3200 tex braided rods.

**Figure 8 materials-16-02459-f008:**
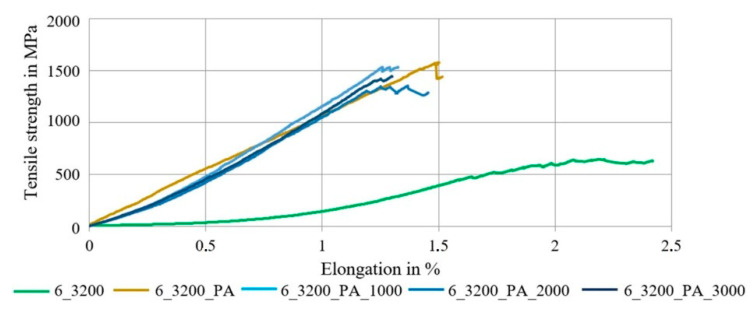
Influence of the pre-tension on tensile properties of impregnated 6 × 3200 tex braided rods.

**Figure 9 materials-16-02459-f009:**
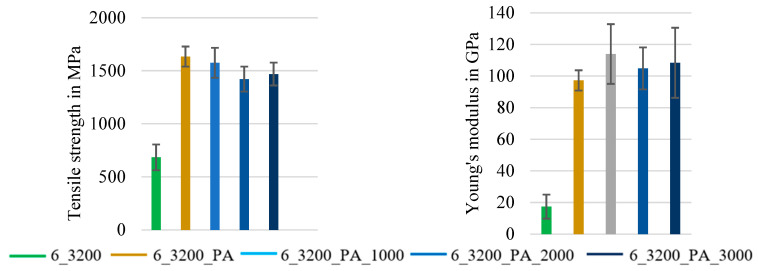
Tensile properties of 6 × 3200 tex braided rods with and without pre-tension.

**Figure 10 materials-16-02459-f010:**
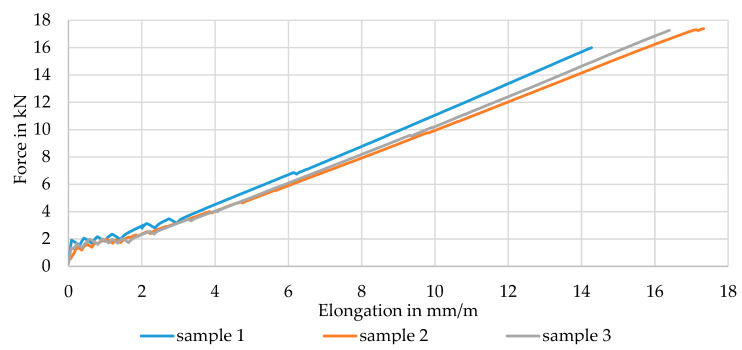
Tensile behavior of CRC-specimens (left) and tested specimen after failure (right).

**Figure 11 materials-16-02459-f011:**
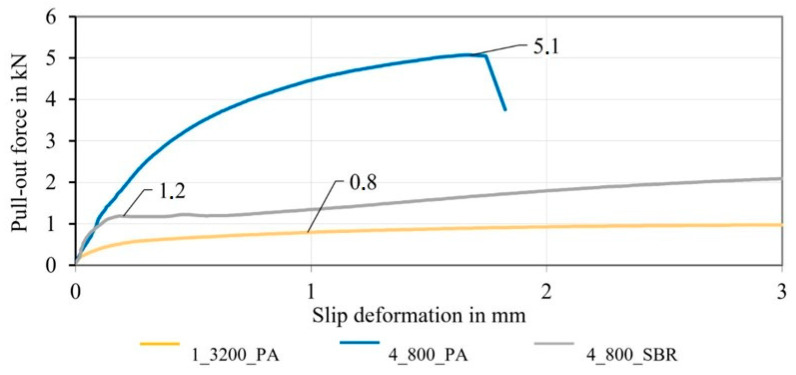
Bond behavior of impregnated braided rods (3200 tex) and different impregnation agents.

**Figure 12 materials-16-02459-f012:**
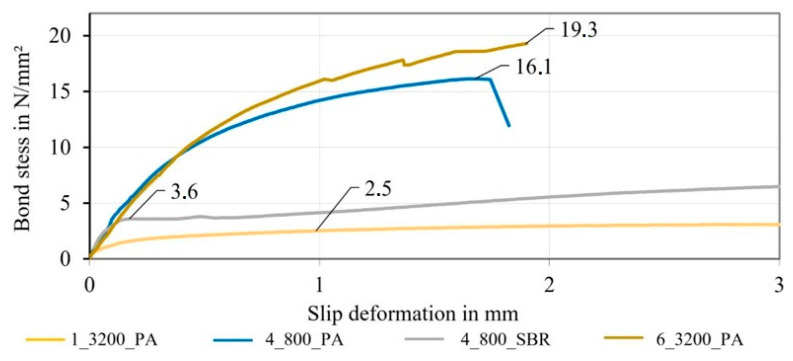
Bond stress of different braided rods.

**Table 1 materials-16-02459-t001:** A brief overview of the literature and the state of the art.

Topic	State of the Art	References
Basic properties of carbon-reinforced concrete (CRC)	Use of carbon filament yarns with smooth surface structure; use of warp knitted grid-like structures with low rod count; high tensile properties; minimal structural elongation for initial load transfer with concrete matrix	[4,5,7,9,10,11,12,27]
Bond properties of CRC	Mainly adhesive bond; no dominant form fit or shear bond; yarn pull-out due to insufficient bonding properties; increased bond lengths for load transfer	[8,13,14,15,16,17,18,19,21,22,23,24,28]
Profiled textile reinforcement structures	Modified yarn structure by twisting and cabling; increased inner bond by fiber friction; no sufficient outer bond; surface profiling of thick rebars through additive/subtractive methods, reduced shear resistance and material efficiency; profiling through reshaping with defined bond properties	[18,20,21,23,25,26]

**Table 2 materials-16-02459-t002:** Characteristics of the used carbon filament yarn [29].

Properties of the Carbon Fiber Heavy Tows						
Product Name	Filaments in k	Count in Tex	Density in g/cm³	Area in mm²	Tensile Strength in MPa	Young’s Modulus in Gpa
Tenax^®^-E HTS45 E23	12	800	1.77	0.45	1632	219
Tenax^®^-E STS40 E23	48	3200	1.77	1.81	1687	232

**Table 3 materials-16-02459-t003:** Overview of the braided rod structures.

Overview of Dry Braided Rod					
Rod Type	Sample Designation	Count in Tex	Braiding Structure	Pitch Length in mm	Illustration
Braided rod	4_800	4 × 800	4-strand flat braid	120	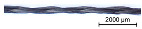
	6_3200	6 × 3200	6-strand flat braid	120	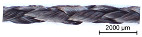

**Table 4 materials-16-02459-t004:** Characteristics of the impregnation agents [31,32].

Overview of the Impregnation Agents				
Product Name	Characteristics	Base Material	Solid Content in %	Linking Temperature in °C
TECOSIT CC 1000 (CHT Germany GmbH)	Aqueous polymer dispersion	Polyacrylate (PA)	47 ± 1	160
Lefasol VL 90 (Lefatex Chemie GmbH)	Styrene-butadiene dispersion	Styrene-butadiene rubber (SBR)	50 ± 1	130–160

**Table 5 materials-16-02459-t005:** Overview of the tested rod architectures (common roving and braided rods) with impregnation.

Overview of Samples with Impregnation					
Material Type	Sample Designation	Rod Count in tex	Impregnation Agent	Polymer Content in Mass %	Pre-Extension Force in N
Impregnated roving as reference					
Impregnated roving	1_3200_PA	1 × 3200	PA	~22	2
Impregnated braided rods with low count					
Impregnated braided rod	*Variation in impregnation agent*				
	4_800_SBR	4 × 800	SBR	~22	2
	4_800_PA	4 × 800	PA	~22	2
	*Variation in solid content*				
	4_800_PA_18	4 × 800	PA	~18	2
	4_800_PA_15	4 × 800	PA	~15	2
	4_800_PA_12	4 × 800	PA	~12	2
	4_800_PA_10	4 × 800	PA	~10	2
Impregnated braided rods with high count					
Impregnated braided rod	6_3200_PA	6 × 3200	PA	~22	20
	*Variation in Pre-Extension force*				
	6_3200_PA_1000	6 × 3200	PA	~22	1000
	6_3200_PA_2000	6 × 3200	PA	~22	2000
	6_3200_PA_3000	6 × 3200	PA	~22	3000

**Table 6 materials-16-02459-t006:** Illustrations of the impregnated braided rods and references.

Overview of Braided Rods
Sample Designation	Rod Countin Tex	Illustration
1_3200_PA	1 × 3200	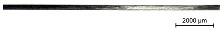
4_800_PA	4 × 800	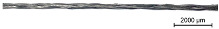
6_3200_PA	6 × 3200	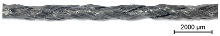

**Table 7 materials-16-02459-t007:** Characteristics of the concrete composition.

Characteristics of the Concrete Matrix			
Product Name	Maximum Grain Size in mm	Compressive Strength in MPa	Flexural Strength in MPa
BMK 45-220-2 [21]	2	≥105	≥11.5
TF 10 CARBOrefit^®^ fine concrete [27]	1	≥80	≥6

**Table 8 materials-16-02459-t008:** Micrographic analysis and geometry parameters of different reinforcement rods.

Micrographic Analysis and Geometry Parameters of Different Reinforcement Rods			
Rod Type	Properties	Values	Illustration
1_3200_PA	Total rod count in tex	3200	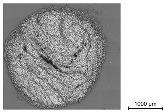
	Comparable diameter in mm	2.0	
	Circumference in mm	6.3	
	Polymer content in mass.-%	~22	
4_800_PA	Total rod count in tex	3200	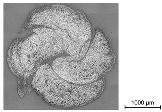
	Comparable diameter in mm	2.0	
	Circumference in mm	6.3	
	Polymer content in mass.-%	~22	
6_3200_PA	Total rod count in tex	19,200	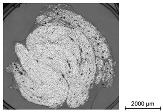
	Comparable diameter in mm	4.5	
	Circumference in mm	14.1	
	Polymer content in mass.-%	~22	

**Table 9 materials-16-02459-t009:** Micrographic analysis of an impregnated reinforcement rod (left—original; right—colored).

Computer-Supported Micrographic Analysis		
Real Microsection	Components	Colored Microsection
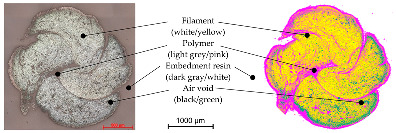		

**Table 10 materials-16-02459-t010:** Evaluation of the tensile tests on CRC specimens.

Sample	First Crack in kN	Crack- Count	Splitting/ Cracking	Longitudinal Crack	Failure Force in kN	Tensile Strength in MPa	Failure Pattern
1	1.91	7	Cracking at failure	at failure	16.00	2946	complete spalling
2	1.37	9			17.40	3204	
3	1.49	9			17.27	3174	
Mean	1.59	8			16.89	3110	

## Data Availability

The data presented in this study are available on request from the corresponding author.

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
