# Peer review of "Improved Tensile and Bond Properties through Novel Rod Constructions Based on the Braiding Technique for Non-Metallic Concrete Reinforcements"

_materials, 2023, doi:10.3390/ma16062459_

Round 1
Reviewer 1 Report
This manuscript focuses the investigations of impregnated surfaces profiled braided rods for an application in concrete reinforcements. Consider, the following few points to attract the reader's attention and improve the article.

Author Response
Dear reviewer,
thank you very much for your helpfull comments and advises. I implemented your recommendations in the paper as stated in the following points (italic):
- Kindly add a Table below the introduction and summarize the critical work as follows.
Table X: A brief overview of the literature.
A brief overview of the literature is given according to the different topics.
- Overall, the article is well written and kindly check for few Typos in few places.
Typos have been corrected.
- Communicating notes can be checked for thoroughness.
Communicatin notes have been checked.
- During the impregnation there is a scope for destabilizing the outcome by the invasion of
Can you elaborate on this.
A paragraph has been added in section 3.2 for the evaluation oft he micrographic analysis.
- Water is added for polymeric dispersion. Has the water quality checks done or specify
what type (E.g., Mineralised, or normal or RO etc. giving their acceptance number).
Normal water is added. This is clarified in section 2.2.
- Pre-extension force is applied, justify with values of input.
The levels of pre-extension force is justified in section 3.1. A short introduction is given in section 2.2
- The values obtained on each case can be at least verified with codes of civil for
construction and later corelated if possible.
The values are verified according to the general building approval of textile reinforcement structures for concrete applications in the sections 3.1, 3.3 and conclusion.
- Yielding stress or safe limits or factor of safety needs to be addressed extensively.
A save limit has been clarified according to the values from the general building approval in the sections 3.1, 3.3 and conclusion.
- When tensile adds in fracture needs to be discussed wherever required.
Fracture and failure patterns have been described where possible and suffiecient, especially in section 3.3.
- Specify the standards of bond tests broadly.
The standards as well as a schematic set up have been added in section 2.4.
- Temperature or pressure is leftover in the discussion. Please throw light upon these two
deciding factors.
The parameters temperature, pressure and relative humidity have been added in section 2.4. The tests were performed according to the norm clima in DIN EN ISO 139.
Reviewer 2 Report
This paper focus on the strength of the braided reinforcement on the concrete and well as the bond strength with the concrete. This paper is well organised and written. The topic is relevant to the journal. However, I have some minor recommendation before publication.
Title should be rephrased to be concise.
Reader will benefit if the first half of the abstract is shortened and the second half of the abstract should also emphases the novelty of this research.
Image in Table 2, table 5 should have a scale in the image to allow reader apricate the size of the braided rod.
Author Response
Dear reviewer,
thank you very much for your helpfull comments and advises. I implemented your recommendations in the paper as stated in the following points (italic):
Title should be rephrased to be concise.
The title has been changed accordingly to “Improved tensile and bond properties through novel rod constructions based on the braiding technique for non-metallic concrete reinforcements“
Reader will benefit if the first half of the abstract is shortened and the second half of the abstract should also emphases the novelty of this research.
The abstract was shortend and the nevelty was emphasised.
Image in Table 2, table 5 should have a scale in the image to allow reader apricate the size of the braided rod.
A scale has been added to the tables.
Reviewer 3 Report
This paper presents braiding technology for the development of novel rod structures from carbon roving with defined yarn properties. The reinforcement structures made from these rods are expected to have significantly improved shear stress transmission in concrete with an adjustable property profile in combination with high tensile strength.
Kindly refer to the following comments for improvement.
1) Include future research of this study in the conclusion
2) Include state-of-the-art-table in the introduction to show the research gap
Specific comments
1. What is the main question addressed by the research?
Find out the effect of a braided rod geometry on the bonding behaviour
2. Do you consider the topic original or relevant in the field? Does it address a specific gap in the field?
Yes
3. What does it add to the subject area compared with other published material?
This paper presents braiding technology for the development of novel rod structures from carbon roving with defined yarn properties, e.g. a defined force-strain behavior, a stretched, shear-resistant anchored yarn structure and surface profiling. The reinforcement structures made from these rods are expected to have significantly improved shear stress transmission in concrete with an adjustable property profile in combination with high tensile strength.
4. What specific improvements should the authors consider regarding the methodology? What further controls should be considered?
The methodology is appropriate.
5. Are the conclusions consistent with the evidence and arguments presented and do they address the main question posed?
Yes
6. Are the references appropriate?
Yes
7. Please include any additional comments on the tables and figures.
Nil.
Author Response
Dear reviewer,
thank you very much for your helpfull comments and advises. I implemented your recommendations in the paper as stated in the following points (italic):
- Include future research of this study in the conclusion
A short paragraph about future research was added in the conclusion.
- Include state-of-the-art-table in the introduction to show the research gap
A brief overview of the state of the art and used literature is given according to the different topics regarded in the paper below the introduction.
Round 2
Reviewer 1 Report
Comments have been addressed as per reviewer suggestions. Article is improved.